# Presence of Flavivirus Antibodies Does Not Lead to a Greater Number of Symptoms in a Small Cohort of Canadian Travelers Infected with Zika Virus

**DOI:** 10.3390/v12020140

**Published:** 2020-01-24

**Authors:** Robert A. Kozak, Lee W. Goneau, Cedric DeLima, Olivia Varsaneux, AliReza Eshaghi, Erik Kristjanson, Romy Olsha, David Safronetz, Stephen Perusini, Christine Frantz, Jonathan B. Gubbay

**Affiliations:** 1Division of Microbiology, Department of Laboratory Medicine and Molecular Diagnostics, Sunnybrook Health Sciences Centre, Toronto, ON M4N 3M5, Canada; 2Department of Laboratory Medicine and Pathobiology, University of Toronto, Toronto, ON M5S 1A8, Canada; 3Public Health Ontario, Toronto, ON M5G 1V2, Canada; GoneauL@dynacare.ca (L.W.G.); cedric.delima@oahpp.ca (C.D.); alireza.eshaghi@oahpp.ca (A.E.); erik.kristjanson@oahpp.ca (E.K.); romy.olsha@oahpp.ca (R.O.); stephen.perusini@oahpp.ca (S.P.); christine.frantz@oahpp.ca (C.F.); 4Government of Canada, Ottawa, ON K1Z 8R9, Canada; Olivia.varsaneux@mail.mcgill.ca; 5Special Pathogens Program, National Microbiology Laboratory, Winnipeg, MB R3E 3R2, Canada; david.safronetz@canada.ca; 6Hospital for Sick Children, Toronto, ON M5G 1V2, Canada

**Keywords:** Zika virus, dengue virus, viremia, antibodies, clinical disease

## Abstract

Zika virus (ZIKV) is a mosquito-borne flavivirus associated with a febrile illness as well as severe complications, including microcephaly and Guillain-Barré Syndrome. Antibody cross-reactivity between flaviviruses has been documented, and in regions where ZIKV is circulating, dengue virus (DENV) is also endemic, leaving the potential that previous exposure to DENV could alter clinical features of ZIKV infection. To investigate this, we performed a retrospective case-control study in which we compared Canadian travellers who had been infected with ZIKV and had serological findings indicating previous DENV or other flavivirus exposure (*n* = 16) to those without any previous exposure (*n* = 44). Patient samples were collected between February 2016 and September 2017 and submitted to Public Health Ontario for testing. ZIKV infection was determined using real-time RT-PCR and antibodies against DENV were identified by the plaque-reduction neutralization test. The mean time from symptom onset to sample collection was 5 days for both groups; the magnitude of viremia was not statistically different (Ct values: 35.6 vs. 34.9, *p*-value = 0.2). Clinical scores were also similar. Our findings indicate that previous DENV or other flavivirus exposure did not result in greater viremia or a higher illness score.

## 1. Introduction

Zika virus (ZIKV) is a mosquito-borne flavivirus spread primarily by the *Aedes* spp., and is a significant public health concern [1]. According to data from WHO, as of March 2018 there were 71 countries that reported introduction, re-introduction or ongoing transmission of the virus (www.who.int). In November 2018, an outbreak has been reported in India, highlighting the ongoing threat posed by this virus [2]. Moreover, in regions where ZIKV has been reported there are other vector-borne flaviviruses that are also endemic, most notably dengue virus (DENV), likely due to these viruses utilizing the same mosquito species as vectors [3]. Consequently, it has been suggested that previous exposure to DENV may increase the severity of subsequent ZIKV infection [4]. A potential mechanism could be the presence of cross-reactive antibodies against DENV that may result in antibody-dependent enhancement (ADE). In heterotypic DENV infections, this process results in more severe disease due to high concentrations of antibodies that bind, but do not neutralize the virus [5]. It has been hypothesized that ADE causes increased ZIKV replication and possibly more severe disease [4,6,7]. It has also been shown that ZIKV induces activation of cross-reactive B-cells in individuals who were previously exposed to DENV [8].

Experiments involving animal models of ZIKV infection have also supported this observation. Notably, Bardina and colleagues demonstrated an increase in mortality in ZIKV-infected *Stat2*-knockout mice that had been pre-treated with DENV-immune plasma [9]. Interestingly, prior exposure to ZIKV in non-human primates also resulted in higher peak viremia following DENV infection, and the serum from these animals demonstrated ADE of DENV in vitro [10]. However, other studies, using different mice strains, and in non-human primates that had been infected first with DENV, and then followed by a ZIKV challenge, have reported conflicting findings [11,12]. While informative, all studies are limited by the use of laboratory-passaged strains, non-natural hosts (in the case of mouse studies) and an artificial route of infection. It has been well established that re-infection with a different serotype of DENV results in ADE [5]. Collectively, there have been conflicting findings that have suggested that antibodies against DENV can cause ADE during ZIKV infection [13], or alternatively that these antibodies neutralize ZIKV, and may even confer protection [7]. Therefore, it remains unclear if previous DENV exposure is associated with a change in ZIKV disease severity or viral replication. This study investigated the relative viral load as well as the number, and severity of reported symptoms in ZIKV-positive patients with serological evidence of previous DENV or other flavivirus exposure compared to ZIKV-positive patients with no serological evidence of previous exposure.

## 2. Materials and Methods

### 2.1. Study Design

A retrospective case control study was performed on samples collected from symptomatic patients from July 2016 to September 2017. This was performed by Public Health Ontario (PHO) Laboratory, Ontario’s reference microbiology laboratory (Toronto, ON, Canada), together with Canada’s National Microbiology Laboratory (NML), Winnipeg, as these organizations were involved in diagnosing ZIKV in returning travelers from regions where the virus was circulating during the ZIKV epidemic, who presented to a healthcare provider in Ontario.

Detection of ZIKV RNA using the Altona RealStar^®^ RT-PCR assay (Altona Diagnostics, Toronto, ON, Canada) was considered as a definitive diagnosis of ZIKV [14]. In this cohort, 60 patients were confirmed to have ZIKV infection by real-time RT-PCR and also had both ZIKV and DENV plaque-reduction neutralization test (PRNT) assay performed by the NML. Sixteen (26.6%) of the 60, who had serological evidence of previous DENV or other flavivirus exposure, were compared to 44 (73.3%) patients with no serological evidence of DENV or other flavivirus exposure.

A patient was considered to have previous exposure based on the results of their PRNT assays, in accordance with the CDC diagnostic guidelines described by Rabe et al. [15]. Patient demographic and syndromic data were collected using a pre-test screening form and all samples were anonymized. For disease severity a Zika illness score was developed where each reported symptom was given one point, and the sum of these points represented the score for the patient. This work was conducted on de-identified specimens submitted as part of routine clinical testing; hence, ethics approval was not required.

### 2.2. Virus and Antibody Detection

Detection of anti-ZIKV IgM antibodies was performed using the CDC ZIKV MAC-ELISA, an IgM-antibody capture assay, and ZIKV RNA detection in serum was performed using the Altona RealStar^®^ RT-PCR, as described previously [16]. To test for DENV and chikungunya virus (CHKV) RNA, we adapted the assay previously described by Pongsiri and colleagues [17]. The plaque reduction neutralization test (PRNT) was performed on all patient samples. The PRNTs were performed in the manner described previously using the viral strains Puerto Rico and DENV-2 New Guinea C [16,18]. Briefly, various dilutions of patient sera were tested with 100 PFU per well. After a 1-h incubation at 37 °C, the mixtures of virus and patient sera were added to the plates containing monolayers of Vero cells. Following this, double overlays of nutrient agar with neutral red were added to the plates to visualize plaque formation over a 3-day period. The dilutions of sera were started at a screening dilution of 20-fold and progressed in a 2-fold pattern. A 90% or greater inhibition of plaque formation was documented as the endpoint dilution/titer [15]. Patients were considered negative for DENV exposure if their PRNT value was zero.

### 2.3. Statistical Analysis

T-tests were performed using SAS (SAS Institute, Cary, NC, USA). Due to insufficient evidence of unequal variances a pooled 95% confidence interval was used for both Ct values and symptoms. Statistical analysis was performed using SAS University Edition. Figures were generated using Graphpad prism version 6.0.

## 3. Results

### 3.1. Cohort Description

The complete description of the cohort is summarized in Table 1. All 60 patients included in this study were travelers to countries in the Caribbean and South America where ZIKV was circulating during the Zika virus epidemic, and ZIKV infection was confirmed by RT-PCR. There were 16 patients with detectable, PRNT-confirmed antibodies against DENV (Table 2), and 44 patients with no serological evidence of DENV exposure, as determined by both plaque-reduction neutralization testing, with a subset also undergoing ELISA testing. Specific serological testing results for other flaviviruses were not available.

In both groups, Jamaica was the most frequently visited country, although several other countries were visited, all of which were in the Caribbean or South America. The mean (51.3 years vs. 43 years) and median ages (52.5 vs. 44 years) were slightly higher in the cohort of patients with DENV exposure (*p* = 0.03). Notably, the average length of stay differed between the two groups (15 days vs. 11 days); however, this failed to achieve statistical significance (*p* = 0.051). This variation could reflect that differences in reason for travel for the two groups. It is possible that individuals with previous DENV or flavivirus exposure may have been returning to visit friends and relatives, spending longer in the ZIKV epidemic area. However, this information was not reported to our laboratory. Symptoms upon presentation to a healthcare provider (HCP) were reported for all patients; the mean time between symptom onset and sample collection was identical for both groups (5 days). Interestingly, while in both groups the majority had detectable ZIKV IgM (Table 1), it was not detectable in all individuals. For both groups, travel history to other regions where flaviviruses were endemic, or vaccination status against yellow fever virus or Japanese encephalitis virus could not be determined.

### 3.2. Relative Magnitude of Viremia

Viremia has been associated with more severe clinical disease for many viruses [19]. As noted in Table 1, the mean time between symptom onset and sample collection was similar for both groups. This was important as it is known that ZIKV viremia decreases over the course of disease [20,21]. Viremia was determined in each group using the Altona ZIKV PCR assay [16], and as shown in Figure 1, Ct values (35.87 vs. 35.14, *p* = 0.2050) and PFU equivalents (*p* = 0.11) were similar between both groups.

### 3.3. Clinical Presentation

Both cohorts had a similar range of symptoms, with rash being most commonly (93.7%) reported in both groups (Table 3). Fever and arthralgia were also reported at approximately similar frequencies. Anorexia, nausea and vomiting were not noted in either group. Interestingly, respiratory symptoms were reported in a small subset of each group, and while this has been reported before in patients infected with ZIKV [22], we cannot rule out that these individuals had a co-infection with a respiratory virus. In terms of total number of symptoms reported, individuals with serological evidence of DENV exposure had an average symptom score of 3.7, compared to 3.4 in patients who had only been infected with ZIKV (Figure 1). There was no significant difference in mean number of symptoms (*p* = 0.3949). Overall, this suggests that the initial severity of infection did not differ between the two groups.

## 4. Discussion

Despite the fact that ZIKV was first described over 70 years ago [23,24], there is still much that remains to be understood regarding pathogenesis. There is increasing interest in investigating the effect of previous flavivirus exposure on ZIKV pathogenesis, and our data support the studies reported from patients in endemic areas [25,26]. Bernardes-Terzian and colleagues also failed to find an increase in ZIKV viremia in Brazilian patients who had serological evidence of previous dengue infection. Additionally, analysis of cytokines noted only modest differences, suggesting disease pathogenesis would be similar. A limitation of their study was the fact that it was done amongst patients from an epidemic area, and the authors did not perform PRNT assays, thereby making it difficult to determine if the antibodies detected in patients were due to DENV or ZIKV [25]. In our cohort, exposure was determined by PRNT, and thus we could more confidently determine who had previous exposure to DENV, although it should be noted that exposure to other flaviviruses could not be entirely ruled out. Additionally, a more recent study by Santiago and colleagues also reported no increase in viral load associated with serostatus [26]. The symptoms associated with ZIKV have been well defined, with rash, fever, arthralgia and conjunctivitis being commonly reported [27]. We did not find significant difference in the number of symptoms reported between the two groups, which is in agreement with Bernardes-Terzian and colleagues [25]. Similarly, it has been shown in pregnant mothers who were infected with ZIKV that prior DENV exposure did not correlate with clinical severity, nor with abnormal birth outcomes, and one study has suggested that prior DENV exposure may even provide protection [28,29]. Additionally, the ZIKV epidemic has not altered the downward trend in DENV severity and mortality that has been reported, further suggesting that exposure to one virus does not result in more severe disease due to the other [4]. Moreover, although co-infection with both viruses has not been frequently reported, in reports or case-series where it has been, an increase in disease severity has not been noted [30].

There are several limitations to our study. One limitation is the potential for cross-reactivity among flaviviruses, which may confound DENV PRNT—the extent to which this occurred in our cohort could not be determined and is a challenge for all flavivirus serological studies. While collecting baseline serologies would be ideal, this is too logistically challenging in our cohort, but may be considered for future studies. Moreover, it has been suggested that PRNT interpretation criteria may need to be adjusted for DENV-endemic areas [31], and how this should be taken into account for extended travel to a region should also be investigated. Since follow up of patients was not possible it cannot be ruled out that their course of disease lasted longer, or that viral shedding or persistence may have differed between groups. Moreover, as the testing was performed at a reference laboratory, we were unable to compare biochemical or hematological profiles between the two groups and did not have direct interaction with the patient. This also limits our ability to ask follow-up questions regarding symptoms that may not have been included in the requisition form. As an example, it has been suggested that yellow fever vaccination may provide moderate protection against ZIKV [32]. While it is part of the standard questionnaire for sample submission, whether or not the patients in either group had received this vaccine, or had documented infection with yellow fever virus may not always have been asked by the health-care provider, although it could be a potential confounding factor. Furthermore, it is not possible to determine if the individual had travelled previously to regions where other flaviviruses (e.g., Spondweni, Kedougou) are endemic, which could also confound results.

Additionally, our data cannot determine whether previous exposure to DENV or other flaviviruses increased or decreased the likelihood of a patient being asymptomatic following ZIKV exposure. Interestingly, recent studies comparing naïve-NHPs to DENV exposed-NHPs, both groups being challenged with ZIKV virus showed comparable number of asymptomatic animals [33]. Larger cohort studies investigating population serologies must be pursued to investigate this in humans. Additionally, it must be considered that the time between DENV exposure and subsequent ZIKV exposure may also play a role in viremia and disease severity, as has been shown with repeated DENV infections [34].

Collectively, our findings add to the evidence that prior flavivirus exposure does not result in more severe ZIKV infection. This is important in the context of vaccination as understanding the role of pre-existing immunity to related viruses can influence vaccine design and implementation. Additionally, it has important clinical and public health implications in regions where the virus is circulating, and for travelers to those areas.

## Figures and Tables

**Figure 1 viruses-12-00140-f001:**
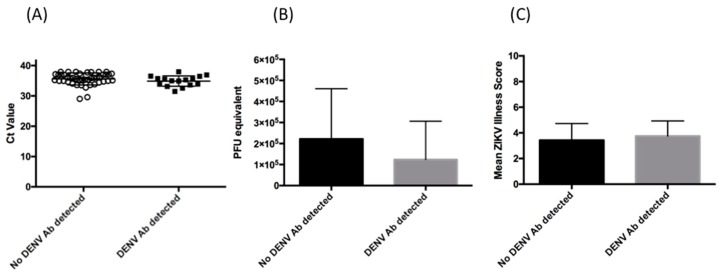
Relative magnitude of viremia and disease severity. (**A**) ZIKV RNA Ct values as a measure of viremia in patients who were DENV PRNT negative (black circles) and those with evidence of DENV or flavivirus exposure, based on DENV PRNT (black squares). The lines represent the mean and error bars represent standard deviation. (**B**) PFU equivalents as in interpolated from Ct values. The bars represent the means, and error bars represent standard deviation. (**C**)Mean Zika illness score for patients. The total number of symptoms for patients with (grey bars) or without (black bars) previous serological evidence of DENV exposure was determined and used to generate a disease severity score. The error bars represent standard deviation. Additionally, as shown in Table 1, the majority of patients were tested for the presence of DENV and CHKV IgM and/or RNA. Overall, this suggests that viremia did not differ between the two groups.

**Table 1 viruses-12-00140-t001:** Characteristics of Zika virus (ZIKV)-positive patient cohorts.

	DENV PRNT Reactive (*n* = 16)	DENV PRNT Non-Reactive (*n* = 44)
Mean age (yrs)	51.9	43
Median age (yrs)	52.5	44
Age range (yrs)	33–88	18–75
% Female	62.5% (*n* = 10)	50% (*n* = 22)
Mean number of days of travel (days)	15 (range 7–33)	11 (range 0–37)
Mean time from last day of travel to symptom onset (days)	3 (range 0–9)	1 (range −8–9)
Mean period from symptom onset to specimen collection (days)	5 (range 0–12)	5 (range 1–13)
ZIKV IgM reactive	87.5% (*n* = 14)	97% (*n* = 43)
DENV PRNT positive	100% (*n* = 16)	0% ^Δ^
% tested for acute DENV *	68% (*n* = 11)	57% (*n* = 25)
% tested for acute CHKV *	63% (*n* = 10)	52% (*n* = 23)

* Patients were tested by RT-PCR or IgM ELISA. No positives were detected amongst tested patients. ^Δ^ All patients were tested.

**Table 2 viruses-12-00140-t002:** Plaque-Reduction Neutralization Titers of patients with serological evidence of previous exposure to flaviviruses, including DENV.

Patient	ZIKA PRNT	DENV PRNT	Fold Difference	Interpretation ^Δ^
1	0	1:20	N/A	Previous dengue virus exposure
2	1:160	1:40	0.25	Flavivirus infection, specific virus cannot be identified
3	1:20	1:40	2	Flavivirus infection, specific virus cannot be identified
4	1:20	1:80	4	Flavivirus infection, specific virus cannot be identified
5	1:40	1:160	4	Flavivirus infection, specific virus cannot be identified
6	1:20	1:160	4	Flavivirus infection, specific virus cannot be identified
7	1:40	1:160	4	Flavivirus infection, specific virus cannot be identified
8	1:320	1:320	1	Flavivirus infection, specific virus cannot be identified
9	1:40	1:1280	32	Flavivirus infection, specific virus cannot be identified
10	0	>1:40	4	Previous dengue virus exposure
11	0	>1:40	4	Previous dengue virus exposure
12	0	>1:40	4	Previous dengue virus exposure
13	0	>1:40	4	Previous dengue virus exposure
14	0	>1:40	4	Previous dengue virus exposure
15	0	>1:40	4	Previous dengue virus exposure
16	1:40	>1:640	16	Flavivirus infection, specific virus cannot be identified

^Δ^ Interpretation is based on CDC diagnostic criteria [15]

**Table 3 viruses-12-00140-t003:** Summary of clinical symptoms reported at presentation.

	DENV or Flavivirus Exposure (*n* = 16)	No Exposure (*n* = 44)
ACHES	18.7% (*n* = 3)	2.3% (*n* = 1)
ANOREXIA	0% (*n* = 0)	2.3% (*n* = 1)
ARTHRALGIA	56.3% (*n* = 9)	43.2% (*n* = 19)
CHILLS	12.5% (*n* = 2)	4.5% (*n* = 2)
CUNJUNCTIVITIS	31.3% (*n* = 5)	20.5% (*n* = 9)
FATIGUE	25% (*n* = 4)	15.9% (*n* = 7)
FEVER	43.8% (*n* = 7)	61.4% (*n* = 27)
GASTROENTERITIS	6.3% (*n* = 1)	6.8% (*n* = 3)
HEADACHE	37.5% (*n* = 6)	31.8% (*n* = 14)
LYMPHADENITIS	6.3% (*n* = 1)	9% (*n* = 4)
MALAISE	6.3% (*n* = 1)	2.3% (*n* = 1)
MYALGIA	25% (*n* = 4)	25% (*n* = 11)
NAUSEA/VOMITING	6.3% (*n* = 1)	4.5% (*n* = 2)
NEUROLOGICAL	0%	6.8% (*n* = 3)
WEAKNESS	0%	2.3% (*n* = 1)
RESP. SYMPTOMS	31.3% (*n* = 5)	9.1% (*n* = 4)
RASH	93.7% (*n* = 15)	93.7% (*n* = 41)
OTHER	12.5% (*n* = 2)	2.3% (*n* = 1)

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
