# Peer review of "Presence of Flavivirus Antibodies Does Not Lead to a Greater Number of Symptoms in a Small Cohort of Canadian Travelers Infected with Zika Virus"

_viruses, 2020, doi:10.3390/v12020140_

Round 1

Reviewer 1 Report

This brief report tries to analyze the impact of previous DENV infection on the severity of Zika infection in a small cohort of 16 human patients.

I have two main issues that must, in my view, be addressed.

1) It is not clear to me if the claim of "prior DENV infection" can be substantiated. It might simply be the case of cross-reactive ZIKV antibodies affecting the DENV PRNT assay.

Was the diagnosis of DENV infection known before any diagnosis of ZIKV infection? If so then this must be better clarified in the manuscript.

If not than I am afraid that the title/conclusion is not appropriate.

2) The title suggests broad implications that are not justified by the small cohort. Provided that issue 1) is addressed and that previous DENV infection is confirmed, I very strongly suggest to indicate "16 patients" in the title. This will allow the reader to correctly evaluate the impact of the study.

Should previous DENV infection not be confirmed, in my opinion the manuscript can only be accepted if "pre-existing" is removed from the title and the tone of the whole manuscript is changed accordingly.

Reviewer 2 Report

The manuscript provides a well-written report on observations that the presence of pre-existing anti-DENV antibodies does not have any significant effect on viremia and clinical severity in patients with symptomatic zika infection. This finding is relevant as concerns has been raised that prior DENV infection may enhance ZIKV infection. But the results of the current study are consistent with a growing body of evidence about lack of enhancement. 

Major comments: 

In the discussion (line 189 onwards), the authors can also mention that another limitation of this (and many other studies) is that the study subjects are limited to patients who reported to the clinic because of symptoms. As most adults infected with ZIKV are asymptomatic, the current study cannot answer the question whether prior DENV infection has any effect on the proportion of people who develop symptomatic infection. 

Minor comments:

Lines 83-105: the citations 14 to 19 need to be as superscript. In addition, the authors should check that the numbering of the citations is correct throughout the manuscript. For example, in line 96, citation  #15 seems to refer to  reference # 17 (description of the PCR method). 

Reviewer 3 Report

Manuscript Review: viruses-638559

Presence of pre-existing antibodies against dengue virus does not affect clinical severity or viremia in patients infected with Zika virus

Robert A. Kozak, Lee W. Goneau, Cedric DeLima, Olivia Varsaneux, AliReza Eshaghi, Erik Kristjanson, Romy Olsha, David Safronetz, Stephen Perusini, Christine Frantz, Jonathan B. Gubbay

Summary:

The manuscript presented by Kozak et. al. attempts to retrospectively assess correlations between previous Dengue virus (DENV) exposure and disease severity in travelers confirmed to be infected with Zika virus (ZIKV) by qPCR. By their measures, the authors detect no differences in either the amount of ZIKV genomic RNA signal in patient serum, nor in the number of disease symptoms suffered by patients when comparing historically DENV-exposed samples to naive individuals. The manuscript attempts to discriminate recently returned Canadian travelers acutely infected with ZIKV as either DENV naive or having previous DENV infection based upon plaque reduction neutralization (PRNT) assay against a laboratory strain of DENV-2. The relative amount of ZIKV RNA in serum collected upon hospital visitation for each set of patients was then compared, finding no difference in ZIKV viremia between groups. The number of symptoms between groups was also compared, with no difference found. While the topic of previous flavivirus exposure affecting severity of disease during subsequent heterologous flavivirus infection (i.e. antibody-dependent enhancement, ADE) is important, this manuscript it its current form does not provide strong enough evidence to make claims for or against ADE in DENV/ZIKV settings in humans.

Overall Opinion:

Unfortunately, several issues and caveats concerning interpretation of the presented data erode confidence in the authors assertion that there is no relationship between prior DENV infection and severity of ZIKV infection. The segregation of patients as DENV naïve or formerly DENV-exposed is based on an assumption rather than solid, confirmed evidence. The measurement of ZIKV load in serum is at a timepoint likely past peak viremia, and the representation of data as Ct value rather than quantified genome equivalents is less appropriate. Comparing simply the number of symptoms between patient sets, rather than weighting symptoms for severity and association with dissemination does not capture a true measure of differences in disease outcomes. For these main shortcomings, I do not recommend this paper for publication in its current form.

Main Comments:

Assumptions regarding prior DENV exposure. The authors assertion that any positive PRNT assay against DENV-2 New Guinea C reflects prior DENV exposure is not adequately justified. The authors should cite references when asserting this parameter in their study, i.e. after statements in lines 106 and 176-177. As admitted in lines 189-191 in the discussion, many different flaviviruses can generate cross-neutralizing antibodies, and indeed it may be that prior exposure was not DENV but rather another flavivirus. Though the authors state that patients had recently returned from Caribbean travel (lines 123-124), that can only account for ZIKV infection, and by no means controls for the geographical site of prior flavivirus exposure. Therefore viruses that might confound PRNT results such as Kedougou virus or Spondweni virus (genetically similar to DENV/ZIKV), or indeed prior African- or Asian-acquired ZIKV exposure may be relevant. Indeed, for at least patients 2 and 8, PRNT titres for ZIKV are equal to or greater than DENV (Table 2), arguing that DENV was not the virus these individuals were previously exposed to.

As prior DENV exposure was the central defining point segregating patients for comparison, evidence of this needs to be much stronger for this paper to be published. As it stands, the reader cannot be sure that patients should be regarded as having prior exposure to DENV, and so any and all further analyses cannot be seen as trustworthy. Do the authors have access to patent histories that relate confirmed diagnoses of DENV infection (i.e. by qPCR) in the past?

Determination of viremia – timepoint and methodology. 5 days post-symptom onset is not the same as 5 days post-ZIKV infection (Table 1) and is likely to be at a point where ZIKV is cleared from human serum (Lessler et al, (2016) Science). Therefore, measurement of ZIKV abundance in patient serum at this timepoint is very likely to be insufficient at relating differences in peak viremia load between groups. Also, could the authors provide a range of days for each cohort that represent time from symptom onset to sample collection, rather than just presenting the mean of 5 days? If there is a lot of variation within groups, presented data becomes even harder to accurately interpret.

Regarding the use of the Altona RealStar qRT-PCR kit for viremia interpretation, the manufacturers of this kit state that it is designed for qualitative rather than quantitative determination of ZIKV infection. For citation 15 referred to in this manuscript, the authors of that article used recombinant ZIKV sequences to generate a standard curve, allowing them to report quantitative genome-equivalents in serum. Why did the authors of this current manuscript elect not to use this quantification method and instead report Ct value? Does the Ct value reflect a standard amount of serum, or a standard amount of input cDNA? The assay presented in Figure 1A could be modified as above to produce more robust, reliable results (however, as this is likely so late post-peak viremia, even reporting of genome equivalents by copy-number qRT-PCR will not provide a clear picture of differences between groups).

Symptoms comparison. The illness scores as described does not seem adequate to reflect differences in disease outcomes between patient groups. Rather than each symptom being given a value of 1 for the purposes of generating Figure 1B, each type of symptom should be weighted to reflect severity and dissemination of the infection to the CNS/reproductive tract/non-typical tissues. “Neurological” and “respiratory” symptoms (Table 3) represent outcomes that differ from the typical course of ZIKV infection, and so should be weighted higher than the more typical and less severe “rash” or “aches.” As it stands, Figure 1B does not reflect meaningful differences between groups. Additionally, it seems that the proportion of infected individuals that present any symptoms at all may be a relevant parameter regarding the contribution of prior DENV exposure (i.e. most people infected with ZIKV develop no symptoms at all – does prior DENV exposure lead to less asymptomatic infection?). Do the authors have access to serum samples from family members and/or colleagues of these patients, or other travelers who display no symptoms to assess cryptic ZIKV infection in the context of naïve or prior DENV infection?

Minor Comments:

The title is far too strong a statement considering the evidence presented in the paper. It should be modified to something along the lines of “Presence of pre-existing dengue virus cross-neutralizing antibodies does not lead to a greater number of symptoms in Canadian travelers infected with Zika virus.”

The reference list does not reflect all in-text references. For example, in line 83, reference 14 does not seem to match. In line 97, there is no Pongsiri and colleagues paper listed at all in the reference list.

The authors continually mention ZIKV endemic areas when relating where patients travelled to. As samples were collected in 2016, it is fair to say that the Caribbean was an epidemic rather than endemic area. As a virus infection cycle becomes stable during establishment of endemicity, once would expect selective pressures to change, possibly affecting antigenic properties, and thus ADE. It is important to make such a distinction.

Round 2

Reviewer 3 Report

The article presented by Kozak, et al has been greatly improved by conservative reinterpretation of the scope and significance of data in light of the inescapable limitations of such a serological study performed at a reference centre remote from direct contact with patients. Reviewer comments have been adequately addressed. The article convincingly conveys that prior exposure of Canadian travelers to DENV (or another cross-reactive flavivirus) does not correspond to increasing numbers of symptoms, nor increased magnitude of serum viral load (at time of serum collection) following infection with ZIKV. Improved referencing that clearly explains the rationale for interpretation (especially regarding CDC diagnostic criteria in ref 15) vastly increases confidence in the data as presented.

Though this study is not the first of it's kind, and though the cohort involved may be small, this work does contribute to the increasing body of evidence that (outside of DENV heterologous serotypes) prior exposure of humans to a flavivirus does not lead to overtly enhanced pathogenesis following infection with a different flavivirus species. This article should be published, with only a few very minor points remaining to be changed.

1) page 3, line 110 - "there" should be "their"

2) page 4, line 135 - in light of the authors' response to reviewers, and changes elsewhere in the manuscript, "endemic" should be changed to "epidemic"

3) page 8, line 189 - "or severity" should be removed. I accept the authors' assertion that (in absence of direct patient consultation that would clarify the severity of each listed symptom for each individual) the number of symptoms an individual has suffered can be interpreted as a rough estimate of disease severity. However, the statement on line 189 can misleadingly be read that the severity of symptoms (rather than disease) was measured in this study. This study measured only the number of symptoms, which was interpreted as severity of disease. Severity of symptoms was not analyzed.

Author Response

Thank you kindly for the reviews of our manuscript. Below is a point-by-point response to your comments.

1) page 3, line 110 - "there" should be "their"

 This change has been made.

2) page 4, line 135 - in light of the authors' response to reviewers, and changes elsewhere in the manuscript, "endemic" should be changed to "epidemic"

This word has been corrected.

3) page 8, line 189 - "or severity" should be removed. I accept the authors' assertion that (in absence of direct patient consultation that would clarify the severity of each listed symptom for each individual) the number of symptoms an individual has suffered can be interpreted as a rough estimate of disease severity. However, the statement on line 189 can misleadingly be read that the severity of symptoms (rather than disease) was measured in this study. This study measured only the number of symptoms, which was interpreted as severity of disease. Severity of symptoms was not analyzed.

Thank you, we have removed these words from the manuscript.